# Safety and health measures compliance during the COVID-19 pandemic for community-based tourism in Nakhon Si Thammarat, Thailand: A cross-sectional descriptive study

Apirak Bumyut[1,2]*, Sasithorn Thanapop[3,4], Ni Made Utami Dwipayanti[5]

1 Department of Environmental Health and Technology, School of Public Health, Walailak University, Thasala, Nakhon Si Thammarat, Thailand, 2 Excellent Center for Dengue and Community Public Health, Walailak University, Thasala, Nakhon Si Thammarat, Thailand, 3 Department of Community Public Health, School of Public Health, Walailak University, Thasala, Nakhon Si Thammarat, Thailand, 4 Research Center of Data Science for Health Science, Walailak University, Thasala, Nakhon Si Thammarat, Thailand, 5 School of Public Health, Faculty of Medicine and Health Sciences, Udayana University, Bali, Indonesia

* apirak.bu@mail.wu.ac.th

**Data Availability Statement:** The submitted manuscript encompasses all data pertinent for

## Abstract

Community-based tourism (CBT) in Thailand faces challenges in adapting to COVID-19 prevention measures. The purpose of the study was to evaluate levels of knowledge, practice, and compliance regarding safety and health measures of the entrepreneur in managing CBT under the Safety and Health Administration (SHA) standard in the new normal situation. A descriptive cross-sectional study was conducted on twenty-one entrepreneurs from three CBTs in three districts in Nakhon Si Thammarat, in the months of February—May 2021. Levels of knowledge and practice were evaluated by questionnaires and compliance level was evaluated by SHA standard checklist. The level of knowledge and practice were categorized sufficient and insufficient, while compliance level was categorized as high or low if scores met or exceeded 80%, based on Bloom's cut-off point. Information on sociodemographic characteristics was also gathered. Fisher's exact test with a 95% confidence level ($\alpha < 0.05$) was used for statistical analysis. The findings revealed that 66.7% and 38.1% of the establishments in the study had sufficient knowledge (Mean ± SD: 46.9 ± 7.2, Max: 55.0, Min: 33.0) and sufficient practice (Mean ± SD: 40.4 ± 9.2, Max: 55.0, Min: 29.0), respectively. In addition, the study found that knowledge level was significantly associated with practice level at a p-value of 0.018. However, compliance level was not related to knowledge and practice. In conclusion, the low level of compliance was due to a lack of understanding and motivation to comply with the standard, and the budget of small establishments in CBT for bringing them up to the SHA standard was quite limited. Therefore, the related organizations should use a variety of strategies to encourage entrepreneurs, such as partnership building and resource support.

public access. Nevertheless, specific datasets are restricted owing to confidentiality obligations related to study participants and regulations set forth by the Walailak University's Human Research Ethics Committee. Researchers interested in these datasets may apply for access by reaching out to the Ethics Committee via email at wu.wuec@gmail.com, provided they fulfill the required criteria.

**Funding:** Yes, The authors disclosed receipt of the following financial support for the research and authorship of this article: The authors received from Walailak University, research grant no. WU-IRG-64-010. The funders had no role in study design, data collection and analysis, decision to publish, or preparation of the manuscript.

**Competing interests:** No.

## Introduction

Thailand has the potential to compete in the tourism industry at 31$^{st}$ out of 140 countries around the world and is ranked 3 in ASEAN, but the assessment found that the 3 indicators that Thailand has the lowest rank in the aspects of health and hygiene, safety and security, and environmental sustainability [1]. Nakhon Si Thammarat province is one of the tourist destinations in southern Thailand. The tourist attractions in Nakhon Si Thammarat province are special and diverse [2], both in terms of ecotourism community and culture. It is a province with beautiful landscapes, a long and famous civilization towards the World Heritage City, that makes it a target for many Thai and foreign tourists. However, after the outbreak of the COVID-19 pandemic, which is a serious contagious disease in March 2020 in Thailand, causing the government declared a state of emergency and lockdown the country [3]. After the outbreak situation has subsided, the government has stimulated spending on tourism in the country by inviting people to travel in the country. This context required the readiness of entrepreneurs to build confidence in people, both tourists and local communities in the standard of care and security to prevent the spread of COVID-19. At present, Thailand has adopted standards in accordance with the Safety and Health Administration (SHA) standards under the project Amazing Thailand Safety and Health Administration [4]. It is a cooperation project of the Tourism Authority of Thailand (TAT), the Ministry of Tourism and Sports, the Ministry of Public Health and government-private partnership in the tourism industry. They are bringing public health measures together with the quality service standards of the establishment to assure tourists that they will have a good experience, happiness and safety from Thailand's products and services. The SHA standard is implemented into 10 types of businesses [4], including the various types of CBT.

CBT is a type of tourism that considers how to maintain the environment, society, and local culture. CBT development plan and management was operated by the community for community benefit [5]. CBT can be effectively managed only through community participation in all development processes [6–9]. Entrepreneurs in Thailand's CBT faced their biggest challenge with the unexpected threat of COVID-19, causing panic among local entrepreneurs due to uncertainty about lockdowns. The local government's restrictions, including the closure of restaurants and bars, directly resulted in revenue losses and business closures, as confirmed by several entrepreneurs [10]. The local health institutes and public health educational organizations should facilitate the community to increase their knowledge and skill in tourism management during the COVID-19 pandemic. Learning experiences by evaluating safety and health need and readiness enhances the local community to gain a better understanding of the health and safety tourism management process according to the various types of establishments. Important determinants that contribute to the SHA standard operation among the community establishments are knowledge and understanding of such standards [11,12], availability of resources arrangement [13], including the readiness of the establishment and community [14,15]. Research on knowledge, practices, and adherence to COVID-19 mitigation measures through safety and health protocols in CBT establishments in Thailand has been limited. This limitation is particularly concerning, as it is essential to address this gap. The knowledge, attitude, and compliance levels of establishment owners are foundational elements for developing establishments that can meet the required standards and effectively contribute to public health and safety in the CBT context.

The goal of the research is to promote and develop CBT management standards that adhere to national SHA standards for safety and hygiene. Based on this goal, the research employed a questionnaire developed based on SHA standards to assess the knowledge, practice, and compliance levels of entrepreneurs in CBTs. This study aims to evaluate entrepreneurs' levels of

knowledge, practice, and compliance with safety and health measures in managing CBT during the new normal situation. The study results can be applied to plan development and promote community tourism establishments for resilience in the next new normal. Additionally, the study will provide depth understanding of the barriers and needs to improve health resilient of CBTs. The study was conducted in Nakhon Si Thammarat province, a city known for its civilization, the growth of ecotourism, and community life in southern Thailand.

## Literature review

### Sustainable community-based tourism

Tourism faces significant vulnerability to the immediate physical impacts of climate change, such as rising sea levels and increasing temperatures. Indirect threats, like shifts in water availability and the spread of diseases, also pose risks. To ensure sustainability in a changing climate, the tourism sector must confront and adapt to these environmental challenges [16]. The criteria for building sustainable tourism consist of sustainable management, socio-economic sustainability, cultural sustainability, and environmental sustainability [17,18]. Establishing CBT in six Taiwanese communities requires the implementation of a thorough strategy, involving monitoring social carrying capacity, addressing environmental conservation, zoning, and planning, and fostering community engagement [18]. CBT recommends an approach for building sustainable CBT in Vietnam, structured around various policy groups. The suggested steps include: 1) Developing guidelines for CBT; 2) formulating policies for planning and developing key community tourist areas; 3) creating policies that link the development of CBT with the protection of natural and cultural environments; 4) establishing policies for tourism management; 5) implementing policies related to human resource development, promotion activities, and the creation of CBT products [19]. CBT in Kampong Grangsil, Jambangan Village, Dampit District, Malang Regency, suggests that the role and involvement of mediators in the participatory development process increased the ability of communities to organize and build sustainable villages [20]. Moreover, earlier studies demonstrated that CBT could enhance climate resilience. For instance, CBT initiatives in Bhutan improved the climate resilience of communities by upgrading houses for homestays. This could potentially decrease households' dependence on agriculture, which is frequently susceptible to climate change impacts like irregular rainfall, windstorms, floods, and the emergence of new pests [21]. CBT in northern Pakistan's Khyber region has shown increased resilience to climate change, attributed to an active community-based governance system, enhanced practices, improved access to social services, disaster preparedness, and heightened awareness of climate change [22].

### Context of CBT in Thailand

CBT in Thailand focus on environmental, social, and cultural sustainability. CBT is community-driven, managed by the community, and benefits both the community and visitors [23]. The 2nd National Tourism Development Plan emphasizes integrating mainstream tourism with CBT to ensure equitable income distribution. Thailand has specific criteria for CBT development, covering sustainable management, benefit distribution, cultural heritage conservation, environmental management, and service safety [24]. In 2020, there were 1,180 CBTs in Thailand, contributing significantly to the economy. The report highlights the importance of CBT for local employment and income distribution to communities [25]. Like other tourism services, CBT experienced the impact of the COVID-19 pandemic. The tourism revenue for the year 2021 in 20 CBT models under the Designated Areas for Sustainable Tourism Administration (DASTA) declined by 40.30% compared to 2020, which had already seen a decrease of 46.54% from 2019. This substantial decline in revenue is attributed to the

effects of the COVID-19 pandemic, leading to nationwide lockdowns. As a result, many areas did not generate additional income from tourism from January 2021 until the end of September 2021 [26].

## Managing COVID-19 pandemic in the context of managing community-based tourism

Managing the COVID-19 pandemic in the context CBT involves numerous CBT initiatives adapting to challenges in their business operations. In southern Malang, Indonesia, CBT recommends that increasing the intention to visit can be achieved by providing diverse information, minimizing the risk of COVID-19 exposure, and improving attitudes toward risk. These measures are seen as beneficial for fostering the development of a sustainable coastal ecosystem management plan [27]. The research conducted in CBT in Peru reveals that the eco-tourism system effectively adjusted to the immediate impacts of COVID-19, attributing this adaptability to the community's adept self-organization skills and cultural strengths. The research suggests that fostering sustained social networks and implementing effective risk management strategies are crucial for reinforcing the community's ability to withstand challenges posed by both COVID-19 and climate change [28]. The study on eco-tourism in Ghana's Savannah region proposes strategies to enhance resilience in the tourism industry. Key recommendations include diversifying the local economy to reduce reliance on tourism, promoting domestic tourism during crises and travel restrictions, and implementing immediate measures such as stimulus packages to support local enterprises impacted by events like the COVID-19 pandemic [29]. CBT operations in Brunei displayed resilience in response to the challenges brought about by COVID-19. Their adaptive strategies included diversifying offerings, leveraging local resources, and integrating technology like virtual engagements and online promotions [30]. The study in Marakau Village, Sabah, North of Borneo founded that successful post-COVID-19 management of community-based ecotourism necessitates a holistic approach, encompassing human and natural resource management, strategic planning, collaboration, and adaptability for sustainability and success [31]. The study in CBT of the Andaman Coast of Thailand showed that key measures addressed included the characteristics of vaccinated tourists, ensuring community readiness by regularly checking service staff for COVID-19, implementing renovations, and designing pandemic-appropriate tourism activities. Public relations efforts aimed at enhancing the community's positive image and community development aligned with the SHA standards [32]. The SHA project in Thailand is a collaborative effort involving multiple government entities and private stakeholders to encourage businesses to adapt to the "New Normal" during the COVID-19 pandemic. The SHA certification signifies the preparedness of tourism entrepreneurs in preventing COVID-19 transmission, building trust among customers [33–35].

## Materials and methods

### Study area

This research collected data from entrepreneurs and assessed establishments identified as part of CBT in three areas that operated during the COVID-19 pandemic and were recommended by the TAT Nakhon Si Thammarat Office. These areas include a rural fishing community in Tha Sala District, a suburban orchard community in Mueang District, and a rural mountain community in Phrom Khiri District, Nakhon Si Thammarat Province, Thailand. The types of establishments participating in CBTs in this study consisted of recreational activities, food establishments, and homestays, all managed by community members.

## Study design

Based on the objective of the studies to assess the current level of knowledge, practice, and compliance of the entrepreneurs in CBTs, the cross-sectional design was used [36]. A cross-sectional descriptive study was conducted in three CBT areas from February to May 2021. The total number of establishments in CBT for the rural fishing community, suburban orchard community, and rural mountain community were 11, 8, and 14, respectively. The study participants were selected based on the inclusion criteria, which include (1) establishment owner; (2) 20 years old and above; (3) at least one year of experience in managing an establishment from the date of joining the CBT, and (4) willingness to participate in the study. There were 21 entrepreneurs participating in the study, which included 5 entrepreneurs from rural fishing communities (45.5% of the community's entrepreneurs), 7 from suburban orchard communities (87.5% of the community's entrepreneurs), and 9 from rural mountain communities (64.3% of the community's entrepreneurs). The survey questionnaire included two parts: (1) assessing the entrepreneur's knowledge and practice of COVID-19 prevention measures and (2) evaluating the entrepreneur's compliance with COVID-19 prevention measures. The research instruments used were questionnaires and the SHA standard checklist [37].

## Research instruments description

The questionnaire if developed based on the SHA standards to measure the level of knowledge and practice and the level of compliance. The content validity of the research tool was examined by three experts, all of whom had an IOC value > 0.6. The reliability of the instrument was assessed using Cronbach's alpha coefficient, which was found to be 0.7.

Knowledge and practice levels on eleven measures of COVID-19 prevention was self-assessment of entrepreneurs using the questionnaire (adapted from the SHA standard measure) [37], where each question was divided into a 5-point scale of knowledge and practice levels. Each measure had a scale with very good knowledge and practice (5 points), good knowledge and practice (4 points), moderate knowledge and practice (3 points), low knowledge and practice (2 points), and very low knowledge and practice (1 point).

**Compliance with measures of COVID-19 prevention.** The assessment of compliance with COVID-19 prevention measures used the SHA standard checklist, which categorizes the guidelines into 10 types of establishments. Each establishment had a slightly different practice according to the context of the establishment type. The assessment results were measured as compliance (1 point) or non-compliance (0 point). The checklist was used by researchers and research assistants who pass training about the criteria of the SHA standard.

## Data analysis

Descriptive statistics (frequency, percentage) were used to summarize knowledge, practice, and compliance data. Total scores of knowledges and practice were categorized into 2 levels, which included sufficient knowledge/practice and insufficient knowledge/practice, and total scores of compliances were categorized in to 2 levels, which included a high level of compliance and a low level of compliance. Entrepreneurs' knowledge, practice, and compliance levels, using Bloom's cut-off point, as sufficient/high if the score was greater than or equal to 80% [38,39].

Fisher's exact test with a 95% confidence level was used to analyze the association between sociodemographic characteristics and the association between knowledge level, practice level, and compliance level. IBM SPSS Statistics version 13 was used for data analysis.

### Ethics statement

This study followed the declaration of Helsinki–ethical principles, and ethical approval was obtained from the Walailak University Ethics Committee (WUEC-21-012-01). Informed consent was obtained from all participants by signing a written document before data collection, and the research procedures fully complied with the Human Ethical Standard for Research of Walailak University.

## Results

### Sociodemographic characteristics

Entrepreneurs in CBT from all three communities (n = 21) who participated in the study were entrepreneurs who were willing to participate and who were involved in the development and implementation of CBT. As shown in Table 1, the research included entrepreneurs from rural fishing communities (n = 5, 23.8%), suburban orchard communities (n = 7, 33.3%), and rural mountain communities (n = 9, 42.9%). The type of establishment in CBT of the study consisted of recreational activity (n = 10, 47.6%), restaurants (n = 5, 23.8%), and homestays (n = 6, 28.6%).

The results showed that most entrepreneurs were women (76.2%) and 57.1% of entrepreneurs age were 41–60 years old. The majority of entrepreneurs had higher education (61.9%) and were traditional people, who lived in the community for more than 40 years (52.4%). In contrast, the majority of entrepreneurs were traditional, and establishing a CBT operation was novel for them. In the study, 52.4% of entrepreneurs had been in business for less than 5 years, followed by 33.3% who had been in business for 6 to 10 years. Entrepreneurs in CBT were different from other entrepreneurs because most of the entrepreneurs had main occupation besides entrepreneurs in CBT consist of agriculturist (23.8%), merchants (28.6%), and others (23.8%). Although most of the entrepreneurs in CBT were new entrepreneurs, 71.4% of entrepreneurs were trained in tourism and establishment operations by many organizations such as the Tourism Authority of Thailand, the Ministry of Tourism and Sports, local government organizations, universities, etc. In addition, 57.1% of establishments receive awards related to tourism. The highest percentage of awards receiving was from the rural mountain community (66.7%) as shown in Table 1. Fisher's exact test showed that training was statistically significantly associated with knowing about SHA standards and tourism rewards with a p-value of 0.019 and 0.002, respectively (Table 2).

### Levels of knowledge, practice, and compliance with COVID-19 prevention measures

The study found that sufficient knowledge and sufficient practice of each measure did not indicate that entrepreneurs would be able to correctly comply with the level that they could pass each measure. According to the findings, 66.7% and 38.1% of the establishments in the study had sufficient knowledge (Mean ± SD: 46.9 ± 7.2, Max: 55.0, Min: 33.0) and sufficient practice (Mean ± SD: 40.4 ± 9.2, Max: 55.0, Min: 29.0), respectively, but establishments which had a high level of compliance are only 4.8% (Mean ± SD: 5.1 ± 1.7, Max: 9.0, Min: 3.0). Fig 1 shows that the percentage of establishments in the rural mountain community had the highest percentage of sufficient knowledge (88.9%) and sufficient practice (55.6%). The result of Fisher's exact test showed that training of establishment and knowledge level was statistically significantly associated with practice level with a p-value of 0.046 (Table 2) and 0.018 (Table 3), respectively.

**Table 1. Sociodemographic characteristics of participants.**

| Variables | Percentage | | | |
|---|---|---|---|---|
| | Total (n = 21) | Community | | |
| | | Rural fishing (n = 5) | Sub-urban orchard (n = 7) | Rural mountain (n = 9) |
| **Gender** | | | | |
| Male | 23.8 | 60.0 | 14.3 | 11.1 |
| Female | 76.2 | 40.0 | 85.7 | 88.9 |
| **Status** | | | | |
| Single | 19.1 | 60.0 | 14.3 | 0.0 |
| Married | 66.7 | 20.0 | 85.7 | 77.8 |
| Widowed | 14.3 | 20.0 | 0.0 | 22.2 |
| **Age** | | | | |
| 20–40 | 23.8 | 60.0 | 14.3 | 11.1 |
| 41–60 | 57.1 | 40.0 | 85.7 | 44.4 |
| > 60 | 19.1 | 0.0 | 0.0 | 44.4 |
| **Education level** | | | | |
| Primary school | 19.0 | 20.0 | 28.6 | 11.1 |
| Secondary school | 4.8 | 0.0 | 14.3 | 0.0 |
| High school | 14.3 | 20.0 | 0.0 | 22.2 |
| Higher education | 61.9 | 60.0 | 57.1 | 66.7 |
| **Residence period (years)** | | | | |
| < = 20 | 33.3 | 40.0 | 42.9 | 22.2 |
| 21–40 | 14.3 | 20.0 | 0.0 | 22.2 |
| > 40 | 52.4 | 40.0 | 57.1 | 55.6 |
| **Operation period (years)** | | | | |
| < = 5 | 52.4 | 40.0 | 85.7 | 33.3 |
| 6–10 | 33.3 | 60.0 | 14.3 | 33.3 |
| > 10 | 14.3 | 0.0 | 0.0 | 33.3 |
| **Main occupation** | | | | |
| Agriculturist | 23.8 | 0.0 | 42.9 | 22.2 |
| Merchants | 23.8 | 20.0 | 28.6 | 22.2 |
| None | 28.6 | 40.0 | 14.3 | 33.3 |
| Others | 23.8 | 40.0 | 14.3 | 22.2 |
| **Tourism standard training** | | | | |
| Yes | 71.4 | 80.0 | 42.9 | 88.9 |
| No | 28.6 | 20.0 | 57.1 | 11.1 |
| **Tourism awards** | | | | |
| Yes | 57.1 | 60.0 | 42.9 | 66.7 |
| No | 42.9 | 40.0 | 57.1 | 33.3 |
| **Knowing about SHA standard** | | | | |
| Yes | 42.9 | 80.0 | 0.0 | 55.6 |
| No | 57.1 | 20.0 | 100.0 | 44.4 |

Fig 2 showed COVID-19 prevention measures that all establishments had the same criteria. Providing wash basins with sanitizer before entering establishments (Measure 5) (90.5%), implementing an adequate air ventilation system (Measure 8) (85.7%), and implementing a safe payment (Measure 10) (81.0%) were the measures that the majority of entrepreneurs could comply with. In contrast, the three majority measures that most of the

**Table 2. The association between training, knowing about SHA standards, tourism rewards, and practice level using a Fisher's exact test.**

| Variables | Knowing about SHA standard | | p-value | Tourism awards | | p-value | Practice level | | p-value |
|---|---|---|---|---|---|---|---|---|---|
| | Yes | No | | Yes | No | | Sufficient | Insufficient | |
| **Tourism standard training** | | | | | | | | | |
| Yes | 9 (42.9%) | 6 (28.6%) | .019* | 12 (57.1%) | 3 (14.3%) | .002* | 8 (38.1%) | 7 (33.3%) | .046* |
| No | 0 | 6 (28.6%) | | 0 | 6 (28.6%) | | 0 | 6 (28.6%) | |

* Statistically significant ($p \leq 0.05$).

entrepreneurs were unable to comply with were communicating and providing knowledge on COVID-19 through various channels (Measure 11) (0.0%), recording employees' travel history (Measure 3) (9.5%), and allowing only service recipients who wore masks (Measure 4) (19.0%).

Fig 3 showed that almost all of each community's compliance levels for each measure did not relate to knowledge and practice levels because compliance was lower than knowledge and practice levels. The sub-urban orchard community outperformed other communities in terms of knowledge because the community had sufficient knowledge of up to 9 measures, but the sufficient practice was the lowest. The rural fishing community performed outstandingly in terms of sufficient practice (6 measures) and a high level of compliance (3 measures).

Finally, the results of the entrepreneurs' compliance with SHA measures revealed that none of the establishments were SHA certified because the SHA standard defined that an establishment must pass all of the criteria in order to be SHA certified. However, when the compliance level was divided into two levels to reflect the entrepreneurs' ability to comply with the SHA standard, it was discovered that only 4.8% of entrepreneurs complied at a high level.

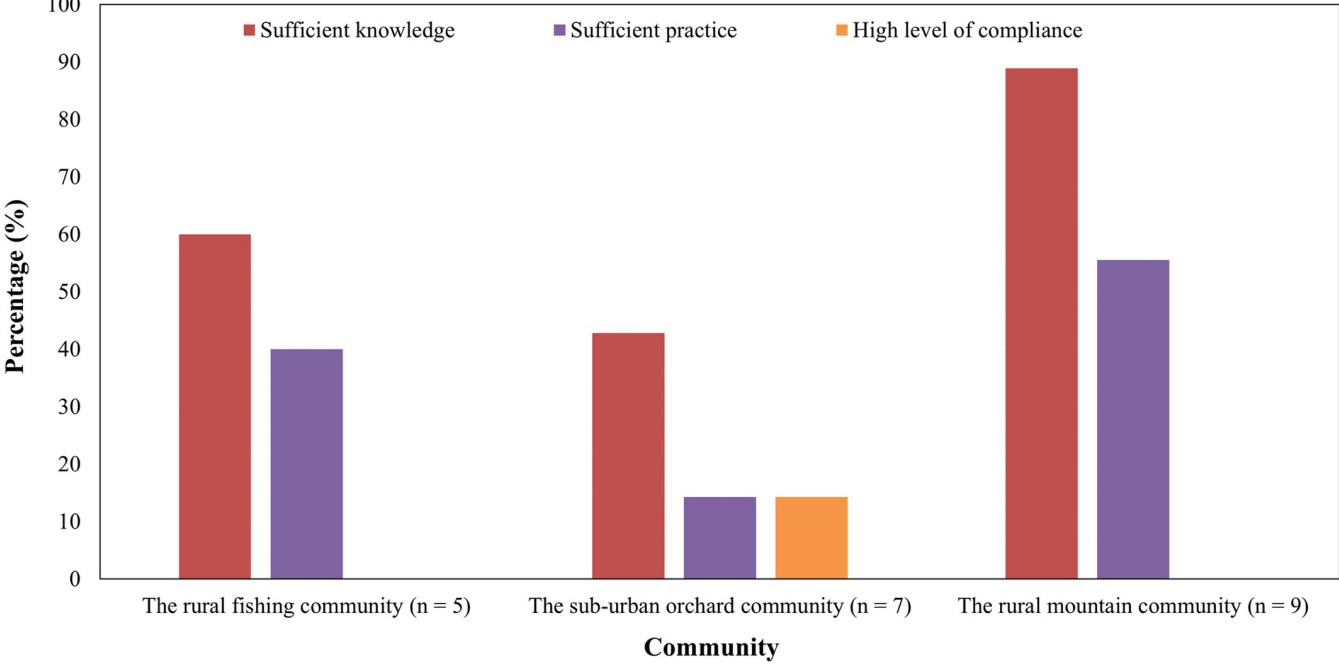

**Fig 1. Percentage of knowledge level, practice level on SHA standard, and compliance level of SHA standard of each community.**

**Table 3. The association between knowledge, practice, and compliance level using a Fisher's exact test.**

| Variables | Practice level | | p-value | Compliance level | | p-value |
|---|---|---|---|---|---|---|
| | Sufficient | Insufficient | | High level | Low level | |
| **Knowledge level** | | | | | | |
| Sufficient | 8 (38.1%) | 6 (28.6%) | 0.018* | 1 (4.8%) | 13 (61.9%) | 1.000 |
| Insufficient | 0 | 7 (33.3%) | | 0 | 7 (33.3%) | |

* Statistically significant ($p \leq 0.05$).

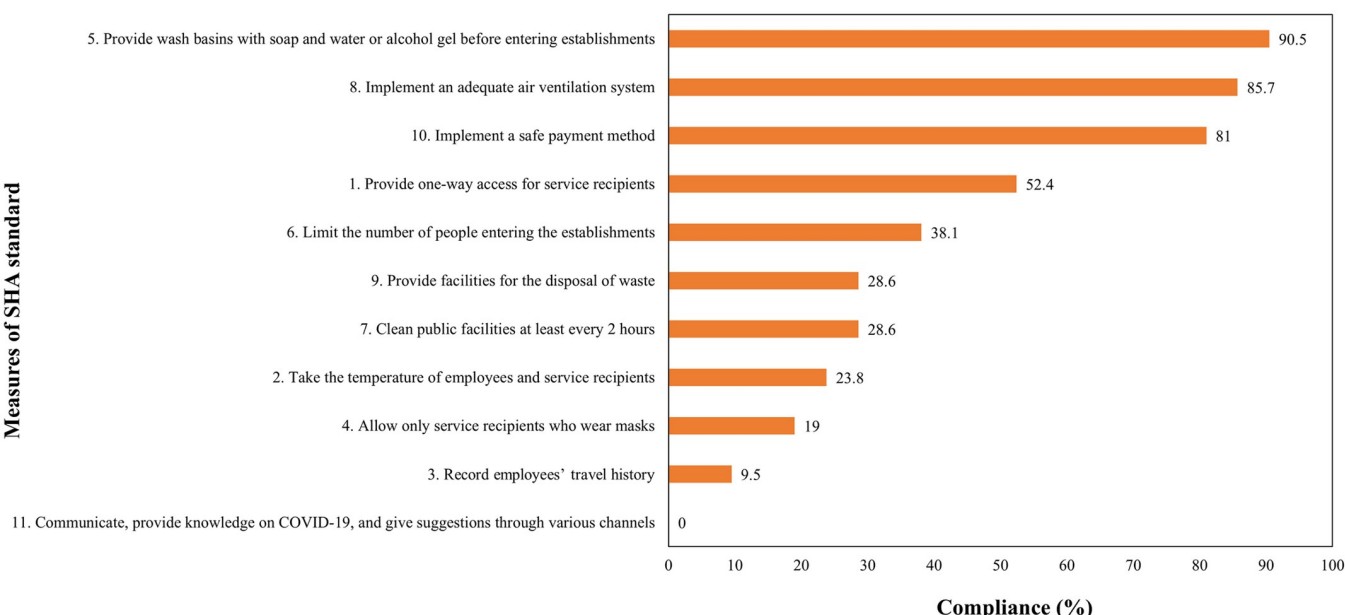

**Fig 2. Compliance percentage with COVID-19 prevention measures.**

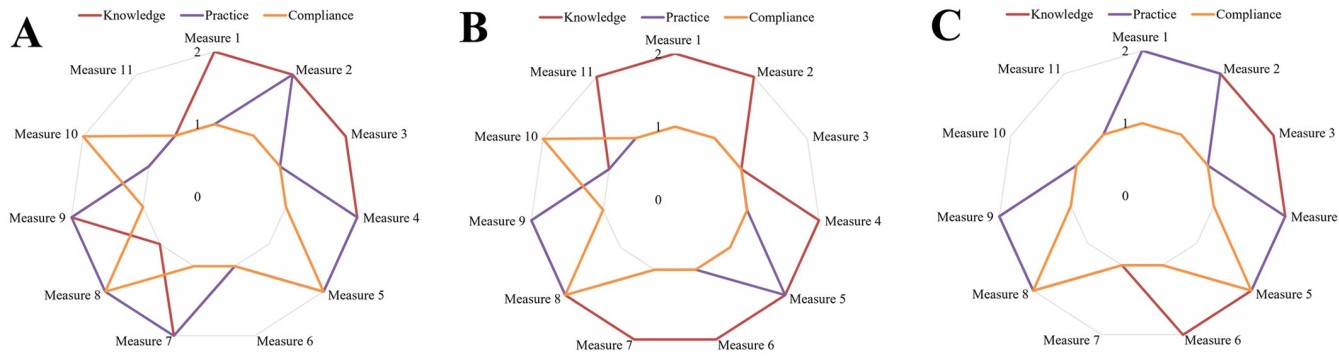

**Fig 3. Knowledge, practice, and compliance levels of (A) the rural fishing community (n = 5), (B) the sub-urban orchard community (n = 7), and (C) the rural mountain community (n = 9) on each of the SHA measures (2 was sufficient/high level, 1 was insufficient/low level).**

## Discussion

### Trained establishment and awareness on the SHA standard

The results showed that most of the establishments that were trained in tourism standards, were affected by Strategy 3 of the twelfth national economic and social development plan (2017–2021) [40]. In addition, CBT development is related with strategy 3 of the second national tourism development plan (2017–2021) [41]. Both of plans clearly defined that government and private organizations must develop essential skills and incentives for personnel in the service and tourism sectors to develop the CBT industry. The positive association between tourism standard training and knowing about SHA standards (Table 2) can be explained by the opportunity for the trained entrepreneurs to connect with the network related to the development of establishments in CBT, thus gaining News and ongoing development support from various organizations. The result was accorded to previous research by Moswete and Thapa (2015) [42] which reported that positive perceptions about CBT development were affected by capacity building and official training by government organizations to entrepreneurs in CBT. In addition, tourism awards or certificates about CBT from various organizations could motivate entrepreneurs to join the training, because entrepreneurs could gain more benefits and credibility, eventually, entrepreneurs who were trained in tourism standards often found development channels to increase the benefits for their own establishments. As a result, entrepreneurs who were trained in tourism standards were more informed about the standards related to the interests of their own area than entrepreneurs who had not received the training [42]. For this reason, the establishment training about tourism standards was significantly related to the practice level. Entrepreneurs who were trained in tourism standards demonstrated awareness of the benefits of establishments, so entrepreneurs must find ways to develop or maintain the benefits of establishments in the COVID-19 pandemic, with SHA standard measures being one of the ways to maintain the benefits of establishments [43].

### Knowledge and practice on the SHA standard by self-assessment

Knowledge level and practice level were significantly associated, but knowledge and practice did not associate with compliance level which was assessed through observation by researchers. Knowledge is associated with practice because entrepreneurs assessed the practice level of themselves following their knowledge. Overall, establishments with sufficient knowledge and practice were in higher percentage when compared with establishments with high level of compliance (Fig 1). The result of the research, in accordance with previous studies on knowledge, attitudes, and practices towards COVID-19 in Southern Ethiopia, reported that the level of COVID-19 prevention knowledge did not correlate with the level of COVID-19 prevention compliance. However, the previous research showed that factors such as occupation, age, and overall knowledge about COVID-19 were related to COVID-19 prevention compliance [44]. The findings indicated that entrepreneurs did not understand the specifics of each measure. The entrepreneurs in this study had sufficient knowledge about COVID-19 prevention not because they understood SHA standards, but because they had been getting information about COVID-19 prevention measures from many sources since the first wave of the pandemic [45,46]. The SHA standard had more detail that strictly than general COVID-19 prevention. Entrepreneurs would hardly comply with the standard if they did not study the SHA website or attend training.

### Compliance with the SHA standard and influencing factors

The previous study found that the certification of public health standards based on the results of the entrepreneurs' self-assessment is not effective because entrepreneurs' level of

understanding is influenced by many media [45]. In addition, previous research found that knowledge and education levels were not the only factors that influenced entrepreneurs to comply with ecotourism measures. Other important factors that influenced entrepreneurs to practice standard measures included social sanction and economic and moral awareness in establishment operations [47]. Moreover, cost and benefit from compliance with standard measures, facilities to support compliance, enforcement capacity and support of government organizations and related organizations were important factors to success in standard measure compliance [48,49].

The reasoning for the result that entrepreneurs' self-assessment of knowledge and practice levels did not correlate with compliance level had two main points.

(1) Lack of understanding in measure details such as Measure 4 that is the entrepreneur informed customers by speech without signs or symbols to request cooperation to wear a mask before entering the service. In addition, entrepreneurs did not strictly follow the measure to recipients because entrepreneurs were afraid of losing customers. This finding was consistent with previous research, which found that customer or tourist needs or pressures are one of the important factors driving an establishment's compliance with environmental standards or projects [50,51]. In addition, the measure limiting the number of people entering the establishments (Measure 6) was one of the measures that were misunderstood by entrepreneurs, which was shown that the entrepreneurs did not specify the maximum limit of customers. The Department of Disease Control suggested for the number of service recipients within the establishment was at least 4 $m^2$ per person or a 50% reduction in customers [52]. Furthermore, the study found that more than 70% of establishments could not comply with Measure 9 (provide facilities for disposal of waste) (Fig 2), where establishments did not provide two bins: general bins and infected bins, which must be placed in a red or black bag, stacked in two layers, and complete with a lid.

(2) Entrepreneurs lacked the motivation to comply with the standard due to the limited number of customers they have. Therefore, the entrepreneurs did not have financial support to comply with the standard and the standard's compliance did not generate more revenue rapidly [53,54]. For example, a measure about cleaning public facilities at least every 2 hours was not carried out. One of the barriers to complying with the measure was the cost of disinfectants as well as a decrease in income due to a decrease in tourist visits. The results reflected that entrepreneurs were not aware of the difference between being SHA-compliant and not SHA-compliant in terms of establishment benefits. During the COVID-19 pandemic, many establishments that did not have SHA standards certification could continue to operate. Therefore, obtaining an SHA standard certification made no difference to an establishment that did not, thus SHA could not persuade the establishment to comply. The issues reflected by entrepreneurs related to happening in Bangkok, The Bangkok Metropolitan Administration (BMA) announced the Order of Temporary Closure of Premises (No. 45) by specifying food or beverage establishments that could open for service after passing the SHA standard inspection [55]. The announcement affected to operation of food or beverage establishments, so entrepreneurs had a very high demand for SHA standards. The event demonstrated that government mandates or regulations, as well as the obvious benefits of establishing standards, are the key determinants of compliance with the standards of the establishment [56,57].

Moreover, measures in the SHA standard were the same measures for both small and large establishments such as SHA measures for accommodation establishments used among homestays, resorts, and hotels, so the benefit of small establishments was higher reduced than large establishments. That is, the benefits gained from standards compliance for small establishments were difficult to beat the cost when compared with larger establishments [58–60].

Measure 5 (90.5% of compliance), Measure 8 (85.7% of compliance), and Measure 10 (81.0% of compliance) were the measures that the majority of entrepreneurs could comply with (Fig 2). Regarding entrepreneurs, they were concerned about self-care by washing their hands with soap or hand sanitizer. Even if the entrepreneurs had to incur additional financial losses, they wanted to pay for the safety of themselves and their families since they believed alcohol gel and soap were effective at destroying SARS-CoV-2. The results were consistent with previous research which stated that risk awareness of the individual and the efficacy of protective measures had resulted in increased compliance with COVID-19 prevention measures [61,62]. In addition, ventilation and secure payment services, as these two measures were relevant to the current context of the communities before COVID-19 facing. Before the COVID-19 pandemic, the majority of establishments in CBT were open-air buildings with natural ventilation and fans, and practically all establishments in Nakhon Si Thammarat province accepted online payments via mobile banking and other applications.

The study reveals that despite the Thai government's promotion of the SHA and SHA Plus standards as crucial tools in combating COVID-19 and supporting tourism, none of the establishments in the research were SHA-certified. Small establishments and CBT faced difficulties in meeting the SHA standard, indicating that government efforts in emphasizing and publicizing standards were insufficient to create awareness and motivate compliance among these entities. Addressing barriers to compliance in the CBT sector requires continuous training, a nuanced understanding of specific measures, and tailored support for smaller establishments. To build resilience in the face of challenges posed by the COVID-19 pandemic and potential future pandemics, the government should focus on initiatives, enforce stringent mandates, provide appropriate channels for learning and meeting standards, and enhance awareness efforts within the CBT industry. However, the study has a limited number of participants because it took place during the COVID-19 pandemic in Thailand, causing the non-operation of the majority of CBT initiatives. Furthermore, specific establishments within the investigated CBTs were temporarily closed. As a result, the study was unable to recruit the participation of all establishments within the CBTs. Therefore, the result is specified for the three areas which not represent all CBTs in Nakhon-Si-Thammarat.

## Conclusion

Entrepreneurs' knowledge and practice levels were sufficient in all CBTs, but compliance was low because entrepreneurs did not clearly understand the measures and lacked the motivation to comply with the standard. The standard could not guarantee or explicitly state that certification would increase revenue or benefits. Furthermore, as CBT establishments were small businesses, their budget for achieving SHA standards was quite limited. As a result, addressing barriers to compliance requires a holistic approach. Efforts should encompass comprehensive training, targeted education on specific measures, economic support from the government and other related organizations, and clear communication of the benefits of compliance. Tailoring initiatives to the unique challenges faced by small establishments in the CBT sector is crucial for fostering a resilient and compliant industry, particularly in the face of evolving challenges in the future such as the COVID-19 pandemic.

## Acknowledgments

The authors express our sincere appreciation to all the entrepreneurs and establishments that participated in this research, forming the backbone of our study on CBT in Nakhon Si Thammarat Province, Thailand. Special thanks to the TAT Nakhon Si Thammarat Office for their valuable recommendations.

## Author Contributions

**Conceptualization:** Apirak Bumyut, Sasithorn Thanapop.

**Data curation:** Apirak Bumyut, Sasithorn Thanapop.

**Formal analysis:** Apirak Bumyut, Sasithorn Thanapop, Ni Made Utami Dwipayanti.

**Methodology:** Apirak Bumyut, Sasithorn Thanapop.

**Project administration:** Apirak Bumyut, Sasithorn Thanapop.

**Visualization:** Apirak Bumyut.

**Writing – original draft:** Apirak Bumyut, Sasithorn Thanapop, Ni Made Utami Dwipayanti.

**Writing – review & editing:** Apirak Bumyut, Sasithorn Thanapop, Ni Made Utami Dwipayanti.

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
