## [Decision Letter · Decision Letter 0]

18 Oct 2023

PONE-D-23-22531Safety and Health Measures Compliance during the COVID-19 Pandemic for Community-Based Tourism in Nakhon Si Thammarat, Thailand: A Cross-Sectional Descriptive StudyPLOS ONE

Dear Dr. Bumyut,

Thank you for submitting your manuscript to PLOS ONE. After careful consideration, we feel that it has merit but does not fully meet PLOS ONE’s publication criteria as it currently stands. Therefore, we invite you to submit a revised version of the manuscript that addresses the points raised during the review process.

We look forward to receiving your revised manuscript.

Kind regards,

Lóránt Dénes Dávid, PhD

Academic Editor

PLOS ONE

Journal Requirements::

"The authors would like to thank Walailak University for financial support for the research and the participants for their cooperation in the study."

"Yes, The authors disclosed receipt of the following financial support for the research and authorship of this article: The authors received from Walailak University, research grant no. WU-IRG-64-010."

"Yes, The authors disclosed receipt of the following financial support for the research and authorship of this article: The authors received from Walailak University, research grant no. WU-IRG-64-010."            

5. We note that Figure 1 in your submission contain [map images which may be copyrighted. All PLOS content is published under the Creative Commons Attribution License (CC BY 4.0), which means that the manuscript, images, and Supporting Information files will be freely available online, and any third party is permitted to access, download, copy, distribute, and use these materials in any way, even commercially, with proper attribution. For these reasons, we cannot publish previously copyrighted maps or satellite images created using proprietary data, such as Google software (Google Maps, Street View, and Earth). For more information, see our copyright guidelines: http://journals.plos.org/plosone/s/licenses-and-copyright.

               1. You may seek permission from the original copyright holder of Figures 1 to publish the content specifically under the CC BY 4.0 license.  

Additional Editor Comments :

Dear Apirak Bumyut, M.Sc,

Our decision: Major revision.

Based on 4 reviews.

Sinderely yours,

Lorant Denes David

Reviewers' comments:

Reviewer's Responses to Questions

**Comments to the Author**

1. Is the manuscript technically sound, and do the data support the conclusions?

Reviewer #1: Yes

Reviewer #2: No

Reviewer #3: Partly

Reviewer #4: Partly

2. Has the statistical analysis been performed appropriately and rigorously? 

Reviewer #1: Yes

Reviewer #2: No

Reviewer #3: Yes

Reviewer #4: Yes

3. Have the authors made all data underlying the findings in their manuscript fully available?

Reviewer #1: Yes

Reviewer #2: No

Reviewer #3: Yes

Reviewer #4: Yes

4. Is the manuscript presented in an intelligible fashion and written in standard English?

Reviewer #1: Yes

Reviewer #2: Yes

Reviewer #3: Yes

Reviewer #4: Yes

5. Review Comments to the Author

Reviewer #1: The study deals with a very current issue, which is particularly prominent in the investigated area. The objective of the research is clear and well defined. At the same time, the scientific foundation of the research is less visible. I recommend a more prominent presentation of the literature review chapter in the study. The methodology of the research is relevant to the realization of the set goals, therefore I accept the results obtained. At the same time, I recommend strengthening the visualization of the results, using figures instead of difficult-to-understand tables. Overall, a good quality study, minor revision is required.

Reviewer #2: This study focuses on the level of knowledge and practices of entrepreneurs during the COVID-19 period, using a questionnaire survey as the research method. Unfortunately, the conclusions of this manuscript are commonplace and can be deduced through intuition without the need for research. For example, the author mentions that the low compliance rate among entrepreneurs is due to a lack of understanding and motivation to comply with standards, which is obvious. Therefore, I believe that this manuscript lacks a unique conclusion or viewpoint to support its publication, and the data from only 33 questionnaires is insufficient to support this conclusion.

Furthermore, this manuscript has some structural and logical flaws that need to be revised, as follows:

The introduction section provides limited discussion on the logic and relationship of the content related to Thai community tourism, COVID-19, and the knowledge and practices of entrepreneurs.

The Ethics Statement is not suitable to appear in the Materials and Methods section; it should be included as formatted text at the end of the manuscript, before the references.

Another flaw of this study is that the chosen research areas do not demonstrate uniqueness or universality. Why did the author select these three regions? Additionally, the author collected only 33 questionnaire responses from these three communities, and it is currently difficult to believe that these data are sufficient to support the conclusion. The author should provide the content of the questionnaire in the supporting documents, as there seems to be a discrepancy between the number and structure of the questionnaire items and the amount of data collected.

The handling of questionnaire data in this study is also preliminary, with little consideration given to factor structure and relationships.

Reviewer #3: The authors selected an actual but challenging topic for their research. Challenging, as the relation of community tourism and pandemic measures is controversial and hard to control the introduction and use of regulations. I regard the paper as original as both the approach and the selected methodology are unique and gap filling.

The structure and the execution of the research is acceptable. My main problem is that in fact there is no literature review in the manuscript, so it is quite a technical paper and hard to understand the context and the need for this particular research.

Some sources are mentioned in the Introduction chapter but is is far not enough. I recommend to write a separate literature review where the different international sources are critically analysed, e.g. https://doi.org/10.21003/ea.V185-08

The implication of the research results are not indicated.

The limitations of the research are not indicated.

Reviewer #4: The topic of the paper is relevant and interesting, but very special. Authors used related literature sources and cited them correctly. Most part of them are from the last few years, but I suggest them to restructure the paper and separated literature review and improve it with further connecting sources.

There were a lot of difficulties and new circumstances based on COVID-19. That was the reason that the purpose of the paper was to evaluate levels of knowledge, practice, and compliance regarding safety and health measures of the entrepreneur in managing CBT under the Safety and Health Administration (SHA) standard in the new normal situation.

In their paper they analysed 34 entrepreneurs from 3 subdistricts – (11) rural fishing-, (8) sub-urban orchard- and (14) rural mountain community – in Nakhon Si Tammarat province in 2021 between February and May.

They used correct methods which helped them to introduce their results. Figures and tables are informative and increase the value of the paper.

I appreciate their results and conclusions which contain both theoretical and practical remark for the future.

6. PLOS authors have the option to publish the peer review history of their article (what does this mean?). If published, this will include your full peer review and any attached files.

Reviewer #1: No

Reviewer #2: No

Reviewer #3: No

Reviewer #4: No

---

## [Author Response · Author response to Decision Letter 0]

12 Jan 2024

We appreciate all of your comments. We have addressed all tasks following your comments.

---

## [Decision Letter · Decision Letter 1]

13 Feb 2024

PONE-D-23-22531R1Safety and Health Measures Compliance during the COVID-19 Pandemic for Community-Based Tourism in Nakhon Si Thammarat, Thailand: A Cross-Sectional Descriptive StudyPLOS ONE

Dear Dr. Apirak Bumyut,

Thank you for submitting your manuscript to PLOS ONE. After careful consideration, we feel that it has merit but does not fully meet PLOS ONE’s publication criteria as it currently stands. Therefore, we invite you to submit a revised version of the manuscript that addresses the points raised during the review process.

We look forward to receiving your revised manuscript.

Kind regards,

Lóránt Dénes Dávid, PhD

Academic Editor

PLOS ONE

Journal Requirements:

Additional Editor Comments:

Minor revision based on reviewers' evaluation.

Reviewers' comments:

Reviewer's Responses to Questions

**Comments to the Author**

1. If the authors have adequately addressed your comments raised in a previous round of review and you feel that this manuscript is now acceptable for publication, you may indicate that here to bypass the “Comments to the Author” section, enter your conflict of interest statement in the “Confidential to Editor” section, and submit your "Accept" recommendation.

Reviewer #2: All comments have been addressed

Reviewer #5: (No Response)

2. Is the manuscript technically sound, and do the data support the conclusions?

Reviewer #2: Yes

Reviewer #5: Yes

3. Has the statistical analysis been performed appropriately and rigorously? 

Reviewer #2: Yes

Reviewer #5: Yes

4. Have the authors made all data underlying the findings in their manuscript fully available?

Reviewer #2: Yes

Reviewer #5: Yes

5. Is the manuscript presented in an intelligible fashion and written in standard English?

Reviewer #2: Yes

Reviewer #5: Yes

6. Review Comments to the Author

Reviewer #2: (No Response)

Reviewer #5: Dear Authors,

I have completed a thorough review of your paper and find it to be relevant and generally aligned with scientific writing norms. However, I recommend major revisions to enhance its overall quality. Below, I have outlined my observations and recommendations:

Title: I suggest shortening the current title.

Abstract: The abstract is well-written. However, I encourage you to remove the structured abstract format (background, methods, results, conclusion). Your abstract lacks a brief description of the methodology used, and consider formatting the results to interpretations for demystifying your findings.

Introduction: To align with scientific norms and standards, I suggest restructuring the introduction section. It should include a general overview of the topic, relevant previous studies, a brief methodology description, brief findings, theoretical contributions, managerial implications, and a clear statement of the research gap you intend to address. The final paragraph should outline the paper structure.

Literature Review: While the literature review is well-structured, I recommend adding additional studies highlighting different perspectives on the topic, especially research conducted in the ASEAN context. Many sentences and statements should be justified in this section.

Methodology: While your chosen research method aligns with the paper's objectives and is well-detailed, it lacks sufficient description and justification for the techniques used.

Results and Discussion: The results are well-presented. In the discussion section, I recommend offering necessary comparisons with previous studies.

Conclusion: The conclusion should succinctly summarize the main contributions of the research in advancing knowledge in the field and the main implications.

References: Consider adding recent references, mainly from reputable journals Q1 and Q2.

7. PLOS authors have the option to publish the peer review history of their article (what does this mean?). If published, this will include your full peer review and any attached files.

Reviewer #2: No

Reviewer #5: No

---

## [Author Response · Author response to Decision Letter 1]

19 Feb 2024

We would like to express our gratitude for the thoughtful feedback provided by the reviewers, which has significantly contributed to the improvement of our manuscript. We have carefully addressed each of the reviewer's comments and suggestions, and I am pleased to inform you that we have incorporated the necessary revisions into the manuscript as follows:

Comment: Title: I suggest shortening the current title.

Response: Thank you for your feedback. While I appreciate your suggestion to shorten the title of my manuscript, "Safety and Health Measures Compliance during the COVID-19 Pandemic for Community-Based Tourism in Nakhon Si Thammarat, Thailand: A Cross-Sectional Descriptive Study," I respectfully prefer to retain the current title as it effectively captures the scope and focus of my research, which has already been considered from four reviewers.

Comment: Abstract: The abstract is well-written. However, I encourage you to remove the structured abstract format (background, methods, results, conclusion). Your abstract lacks a brief description of the methodology used, and consider formatting the results to interpretations for demystifying your findings.

Response: Thank you for your recommendation. I agree with it, and consequently, I have removed the structured abstract format (background, methods, results, conclusion) from the manuscript in lines 23–41. Additionally, I have included more details to provide a brief description of the methodology in lines 26–32.

Comment: Introduction: To align with scientific norms and standards, I suggest restructuring the introduction section. It should include a general overview of the topic, relevant previous studies, a brief methodology description, brief findings, theoretical contributions, managerial implications, and a clear statement of the research gap you intend to address. The final paragraph should outline the paper structure.

Response: Thank you for your valuable feedback. I appreciate your suggestion to restructure the introduction section to align with scientific norms and standards. I want to assure you that I have already incorporated the suggested elements in the introduction, including a relevant previous study (lines 76 - 79), a brief methodology description and standards that will be used the assess the compliance (lines 57 - 64), research gap (lines 68 – 71 and 79 - 84), and managerial implications (lines 90 - 92). Based on your recommendation, a brief methodology description was added in lines 86 – 88.

Comment: Literature Review: While the literature review is well-structured, I recommend adding additional studies highlighting different perspectives on the topic, especially research conducted in the ASEAN context. Many sentences and statements should be justified in this section.

Response: Thank you for your valuable feedback. In response to your suggestion to include additional studies highlighting different perspectives on the topic, especially research conducted in the ASEAN context, we have incorporated three specific examples in the literature review to enhance the diversity of perspectives. These examples cover Community-Based Tourism (CBT) in Kampong Grangsil, Jambangan Village, Dampit District, Malang Regency, Indonesia (lines 112 – 115), CBT in Brunei (lines 153 – 156), and CBT in Marakau Village, Sabah, North of Borneo (lines 156 - 159). 

Comment: Methodology: While your chosen research method aligns with the paper's objectives and is well-detailed, it lacks sufficient description and justification for the techniques used.

Response: Thank you for your suggestion. We have addressed this concern by incorporating references to support the chosen methodology in lines 180 – 181.

Comment: Results and Discussion: The results are well-presented. In the discussion section, I recommend offering necessary comparisons with previous studies.

Response: Thank you for your suggestion. We appreciate your positive assessment of the presentation of our results. In response to your suggestion, we have incorporated comparisons with previous studies in the discussion section. Specifically, we have included a comparison with the research on "Knowledge, attitudes, and practices towards COVID-19: A community survey in Southern Ethiopia," as suggested, in lines 349 - 353. Additionally, we have included references to other relevant previous research to support our discussions:

Lines 328 – 337: Positive perceptions about CBT development were influenced by capacity building and official training provided by government organizations to entrepreneurs in CBT.

Lines 364 – 368: Knowledge and education levels were not the sole factors influencing entrepreneurs' compliance with ecotourism measures.

Lines 405 – 409: The benefits gained from standards compliance for small establishments were challenging to justify in comparison with the costs incurred, especially when compared to larger establishments.

Lines 415 – 417: Increased awareness of risks among individuals and the effectiveness of protective measures have led to improved compliance with COVID-19 prevention measures.

These additional references provide a more comprehensive understanding of the factors influencing compliance and outcomes discussed in our research.

Comment: Conclusion: The conclusion should succinctly summarize the main contributions of the research in advancing knowledge in the field and the main implications.

Response: Thank you for the valuable feedback. In response to your suggestion, I have ensured that the conclusion effectively summarizes the main contributions of the research in advancing knowledge within the field. This information can be found succinctly presented in lines 441 - 445. Additionally, the main implications of the study are summarized in lines 445 - 451. I believe this approach provides a clear and concise overview of the research's significance and its potential impact on the field. Your guidance has been instrumental in refining the conclusion for better clarity and focus.

Comment: References: Consider adding recent references, mainly from reputable journals Q1 and Q2.

Response: Thank you for your recommendation. Your suggestion is essential for improving the quality of the manuscript. Therefore, I have included the original article in Q1, published on August 3, 2023, in Line 342. Furthermore, we aim to ensure the high quality of references in our manuscript by citing high-quality publications as follows: Out of the 62 references in the manuscript, which include 45 research articles, all the research articles in the reference section were categorized into 25 Q1 publications (55.6%) and 9 Q2 publications (20.0%). Additionally, 34 research articles (75.6%) were published between 2019 and 2024.

 We believe that these revisions have significantly strengthened the manuscript and addressed the concerns you raised. We are grateful for the opportunity to improve our work under your guidance, and we trust that the revised manuscript now meets the high standards set by PLOS ONE.

---

## [Editor Report · Decision Letter 2]

21 Feb 2024

Safety and Health Measures Compliance during the COVID-19 Pandemic for Community-Based Tourism in Nakhon Si Thammarat, Thailand: A Cross-Sectional Descriptive Study

PONE-D-23-22531R2

Dear Dr. Bumyut,

We’re pleased to inform you that your manuscript has been judged scientifically suitable for publication and will be formally accepted for publication once it meets all outstanding technical requirements.

Kind regards,

Lóránt Dénes Dávid, PhD

Academic Editor

PLOS ONE
---

## [Editor Report · Acceptance letter]

23 Feb 2024

PONE-D-23-22531R2 

PLOS ONE

Dear Dr. Bumyut, 

I'm pleased to inform you that your manuscript has been deemed suitable for publication in PLOS ONE. Congratulations! Your manuscript is now being handed over to our production team.

Kind regards, 

on behalf of

Dr. Lóránt Dénes Dávid 

Academic Editor

PLOS ONE